# The Efficacy and Safety of Balloon Enteroscopy-Assisted Endoscopic Retrograde Cholangiography in Pediatric Patients with Surgically Altered Gastrointestinal Anatomy

**DOI:** 10.3390/jcm10173936

**Published:** 2021-08-31

**Authors:** Kensuke Yokoyama, Tomonori Yano, Atsushi Kanno, Eriko Ikeda, Kozue Ando, Tetsurou Miwata, Hiroki Nagai, Yuki Kawasaki, Yamato Tada, Yukihiro Sanada, Kiichi Tamada, Alan Kawarai Lefor, Hironori Yamamoto

**Affiliations:** 1Department of Medicine, Division of Gastroenterology, Jichi Medical University, 3311-1 Yakushiji, Shimotsuke, Tochigi 329-0498, Japan; r0760ky@jichi.ac.jp (K.Y.); tomonori@jichi.ac.jp (T.Y.); r1403ie@jichi.ac.jp (E.I.); kozue_ando@jichi.ac.jp (K.A.); tetsurou_miwata@yahoo.co.jp (T.M.); m05069hn@jichi.ac.jp (H.N.); kawasakiyuki1219@gmail.com (Y.K.); tadayamatoday@gmail.com (Y.T.); tamadaki@jichi.ac.jp (K.T.); yamamoto@jichi.ac.jp (H.Y.); 2Department of Surgery, Division of Gastroenterological, General and Transplant Surgery, Jichi Medical University, 3311-1 Yakushiji, Shimotsuke, Tochigi 329-0498, Japan; yuki371@jichi.ac.jp (Y.S.); alefor@jichi.ac.jp (A.K.L.)

**Keywords:** balloon enteroscopy-assisted endoscopic retrograde cholangiography, double-balloon enteroscopy, surgically altered gastrointestinal anatomy

## Abstract

Balloon enteroscopy-assisted endoscopic retrograde cholangiography (BEA-ERC) is useful and feasible in adults with pancreatobiliary diseases, but its efficacy and safety have not been established in pediatric patients. We compared the success rate and safety of BEA-ERC between adults and pediatric patients. This single-center retrospective study reviewed 348 patients (pediatric: 57, adult: 291) with surgically altered gastrointestinal anatomies who underwent BEA-ERC for biliary disorders from January 2007 to December 2019. The success rate of reaching the anastomosis or duodenal papilla was significantly lower in pediatric patients than in adult patients (66.7% vs. 88.0%, *p* < 0.01). The clinical success rate was also significantly lower in pediatric patients (64.9% vs. 80.4%, *p* = 0.014). The rate of adverse events was significantly higher in pediatric patients than in adults (14.2% vs. 7.7%, *p* = 0.037). However, if the anastomotic sites were reached in pediatric patients, the treatment was highly successful (97.3%). The time of reaching target site was significantly longer in pediatric patients than in adult patients. This study shows that BEA-ERC in pediatric patients is more difficult than that in adult patients. However, in patients where the balloon enteroscope was advanced to the anastomosis, clinical outcomes comparable to those in adults can be achieved.

## 1. Introduction

Double-balloon enteroscopy (DBE) [1] enables visualization and treatment of the distal small intestine of patients with biliary disorders and surgically altered gastrointestinal anatomy [2]. Balloon enteroscopy-assisted endoscopic retrograde cholangiography (BEA-ERC) has been reported to be feasible and successful in patients with surgically altered gastrointestinal anatomy [3,4,5,6,7,8,9,10]. Recently, a short-type balloon endoscope with a large working channel has been developed to enable more effective treatments [11,12,13,14,15]. Shimatani et al. [16] reported that the success rate of reaching the target site was 97.7%, and the therapeutic success rate was 97.9%. At our institution, we have performed BEA-ERC for patients who have previously undergone gastrectomy with Roux-en-Y reconstruction (RYG), hepaticojejunostomy (RYHJ), pancreaticoduodenectomy (PD), gastrectomy with Billroth-II reconstruction (B-II), and living donor liver transplantation (LDLT) (Figure 1). Patients who undergo these procedures are sometimes required to undergo BEA-ERC to treat problems such as anastomotic stenoses. BEA-ERC facilitates treatment of these conditions (Figure 2).

Some studies have examined the success rate and safety of BEA-ERC in high-risk patients [17,18,19,20,21]. Hakuta et al. [22] reported the feasibility of BEA-ERC in elderly patients. Reports of BEA-ERC in pediatric patients have increased in recent years. In a nationwide survey in Japan, Kudo et al. [23] reported that pediatric gastrointestinal endoscopies had higher complication rates owing to increasing numbers of patients undergoing endoscopic retrograde cholangiopancreatography (ERCP) and small intestine enteroscopy. The study by Mercier et al. [24] of 15 French and Belgian centers reported a 19% adverse event rate for pediatric patients undergoing ERCP. However, the feasibility and safety of BEA-ERC in pediatric patients have not been established. This study sought to clarify the efficacy and safety of BEA-ERC for biliary disorders in pediatric patients with surgically altered anatomies. The primary end-point was to compare the success rates of reaching the target site and clinical success rates of BEA-ERC in adult and pediatric patients. The secondary aim was to compare the rate of adverse events associated with BEA-ERC in pediatric and adult patients.

## 2. Patients and Methods

### 2.1. Patients

This study was approved by the Institutional Review Board of Jichi Medical University (No. JICHI 24537). This retrospective single-center study included patients with surgically altered gastrointestinal anatomy who underwent BEA-ERC between January 2007 and December 2019 at Jichi Medical University Hospital, Tochigi, Japan. In total, 348 patients (680 sessions) were included. Written informed consent, including confirmation of understanding the use of the data relating to the procedure for research, was obtained from each patient before performing the procedure. The definitions of pediatric and adult patients were patients ≤18 years old and >18 years old, respectively. We exclude the patients with the malignant biliary disorder because the pediatric pa-tients did not have malignancy disease.

### 2.2. BEA-ERC

Before 2015, a short-type double-balloon endoscope with a 2.8 mm channel (EI-530B and EC-450BI5, Fujifilm, Tokyo, Japan) was used at our institution. Thereafter, we performed BEA-ERC using a short-type DBE (EI-580BT with a 3.2 mm channel, Fujifilm, Tokyo, Japan). Endoscopy was performed by expert endoscopists and caregivers. The procedures were performed by K Yokoyama, T Yano, Y Kawasaki, H Hatanaka, and H Yamamoto, who are experts of DBE. All patients underwent BEA-ERC with CO_2_ insufflation. We have not used prophylaxis medicine to prevent pancreatitis. We initiated endoscopic insertion in the left lateral position, which maintains the anteroposterior diameter of the abdomen and assists with abdominal compression, facilitating endoscopic insertion. Intramural injection of indigo carmine was performed to evaluate the afferent loop [25]. After the insertion of the endoscope into the afferent limb, we suctioned the air and intestinal fluid and replaced them with clear water to maintain a clear field-of-view and prevent an increase in abdominal pressure. Upon reaching the target site, we changed the patient from a lateral position to a semisupine position. Biliary cannulation was performed with a cannula (ERCP-catheter, MTW Endoskopie Manufaktur, Wesel, Germany) and 0.025 or 0.035 in guidewire (VisiGlide2, Olympus, Tokyo, Japan). Balloon dilation (REN biliary dilation catheter, Kaneka, Osaka, Japan) was performed for patients with a tumor of the papilla of Vater or a stricture of the anastomosis. We performed the removal of bile duct stones using a basket catheter (Flower basket and LithoCrush, Olympus, Tokyo, Japan) and a balloon catheter (Extraction Balloon, Olympus, Tokyo, Japan). Plastic stents (Through Pass, GADELIUS, Tokyo, Japan) were inserted for drainage of bile duct.

### 2.3. Sedation and Anesthesia

We performed BEA-ERC during intravenous sedation or general anesthesia. We selected the type of anesthesia according to the patient’s general condition and characteristics. General anesthesia was used for most pediatric patients, while intravenous sedation was selected for most adult patients. However, for children whose physiques were comparable to an adult, we selected intravenous sedation. For intravenous sedation, we used midazolam and pethidine or pentazocine.

### 2.4. Evaluation of Procedures

The endoscopic target in pediatric patients was the anastomotic site of the bile duct or small intestine. In all pediatric patients with altered gastrointestinal anatomy, the bile duct had been resected and anastomoses of the bile duct and small intestine were present. The target site in adult patients was the anastomosis or the papilla of Vater. We defined clinical success as the successful biliary cannulation, diagnosis, and biliary drainage and, wherever necessary, subsequent balloon dilation, stone removal, and stenting. We compared the success rates between pediatric and adult patients who had statuses post RYG, RYHJ, PD, B-II, and LDLT. For patients who required multiple endoscopic sessions, the first treatment was included in the analysis. However, if the first treatment was clinically unsuccessful, the first successful subsequent procedure was analyzed. Success in reaching the target site and clinical success in pediatric patients who had an intussusception antireflux valve were defined as accessing this valve and dilating it, respectively. The procedure time was the length of time during which the endoscope was in the patient.

### 2.5. Statistical Analysis

Categorical variables were expressed as numbers and compared with the use of Fisher’s exact test performed with the GraphPad Prism software (GraphPad Software, San Diego, CA, USA). Continuous variables were expressed as the median and interquartile range (IQR) and were compared using Student’s t-test performed using the EZR software (version 1.41, Saitama Medical Center, Jichi Medical University, Saitama, Japan; http://www.jichi.ac.jp/saitama-sct/SaitamaHP.files/statmedEN.html (accessed on 6 April 2021)), which is a graphical user interface for R (version 4.0.3, The R Foundation for Statistical Computing, Vienna, Austria).

## 3. Results

### 3.1. Patients

Patient characteristics are listed in Table 1. Fifty-seven pediatric patients (127 sessions) with an average age of 10.0 ± 4.3 years were analyzed. Among them, 53 underwent LDLT and 4 underwent RYHJ. Biliary atresia was the most common primary disease among the LDLT patients (Table 2). Overall, 291 adult patients (553 sessions) with an average age of 65.9 ± 17.3 years were reviewed. The number of patients who underwent LDLT, RYHJ, RYG, and B-II was 32, 72, 102, and 36, respectively. The most common disease that required RYG was gastric cancer. The type of surgery varied among pediatric and adult patients. General anesthesia was administered to 44/57 (77.2%) pediatric patients and intravenous sedation was given to 282/291 (96.9%) adult patients. Significant differences were observed between pediatric and adult patients with regard to gender, age, type of surgical procedure, indications, and type of anesthesia.

### 3.2. Success Rate and Progress

The success rate of reaching the anastomosis after hepaticojejunostomy, or the papilla of Vater, was significantly lower in pediatric patients than in adults (66.7% vs. 88.0%, *p* < 0.01) (Table 3). The clinical success rate was also significantly lower in pediatric than in adult patients (64.9% vs. 80.4%, *p* = 0.014). Reaching the anastomosis during BEA-ERC was more difficult in pediatric patients. The time needed to reach the target site was significantly longer in pediatric patients than in adult patients (*p* < 0.01). However, if the anastomotic site was reached in pediatric patients, the rate of successful treatment was high (97.3%). In addition, there were no significant differences in clinical success between 2.8 mm and 3.2 mm channels in pediatric and adult cases (31/49 vs. 6/8, *p* = 0.699 in pediatric cases; 164/206 vs. 70/85, *p* = 0.630 in adult cases), respectively. In pediatric patients in whom treatment was not successful, eight involved early percutaneous transhepatic biliary drainage (PTBD) procedures and three involved PTBD after follow-up. In addition, eight patients were treated non-operatively.

Endoscopic insertion was more difficult in pediatric patients owing to their smaller physiques. We analyzed separately pediatric patients with biliary atresia after LDLT at the age of 12 years when their physical changes became stable compared with the endoscopic insertion rates (Table 4). No significant difference was observed in the success rates of reaching the target site in these older pediatric patients compared with younger ones (60.0% vs. 61.5%, *p* = 0.743). The procedure time was significantly shorter in older pediatric patients than in adult patients.

### 3.3. Adverse Events

The rate of adverse events was significantly higher in pediatric patients than in adult patients (14.2% vs. 7.7%, *p* = 0.037) (Table 5). In pediatric patients, cholangitis, mucosal damage, and pancreatitis were the most common adverse events (2.3%). Most patients were treated non-operatively. Although no perforations were diagnosed, one bile duct injury necessitated reoperation for repeat anastomosis. In addition, some pediatric patients presented with fever of unknown etiology. In adults, the most common adverse event was pancreatitis (4.5%), which was severe in two patients (one of the two patients died). No patients had anesthesia-related adverse events.

### 3.4. Intussusception Antireflux Valve

Six adult patients with biliary atresia who had an intussusception antireflux valve in the afferent limb underwent BEA-ERC (Figure 3). Three of the patients were classified with a post-LDLT status. The antireflux valve was created by invaginating the small intestine of the afferent limb to prevent cholangitis in an operation prior to LDLT [26]. However, some reports suggest that the antireflux valve is not effective for preventing cholangitis [27,28]. The indication for BEA-ERC in four patients was cholangitis, possibly owing to (a) a stricture at the hepaticojejunostomy anastomosis or (b) to an intrahepatic bile duct stone.

The antireflux valve makes endoscope insertion very difficult. We were able to pass the endoscope distal to the valve without dilation in one patient for treatment, while the other patients required dilation with a balloon catheter. The anastomosis could not be reached in three patients owing to adhesions of the small intestine.

## 4. Discussion

This study demonstrates the clinical feasibility and safety of BEA-ERC for pediatric patients with surgically altered gastrointestinal anatomy. To the best of our knowledge, this study on BEA-ERC has examined the largest number of pediatric patients (57 pediatric patients who underwent 127 sessions of BEA-ERC). This study showed that the success rate of reaching target site and the clinical success rate were significantly lower in pediatric than in adult patients. The clinical baseline characteristics differed between pediatric and adult patients. Other reasons for the low rate of reaching the anastomosis and clinical success rate may include multiple adhesions and growth of the small intestine as a result of having undergone LDLT in early childhood. At our institution, 75.4% of the primary disease in pediatric patients who underwent BEA-ERC was biliary atresia. Pediatric patients with biliary atresia underwent a Kasai operation shortly after birth. A few years after birth, most patients who had biliary atresia underwent LDLT owing to liver dysfunction. Therefore, most patients at our hospital were classified with the post-LDLT status (71.9%). As patients with biliary atresia needed to undergo multiple operations, which resulted in intra-abdominal adhesions, reaching the anastomosis was difficult in pediatric patients who were classified with the post-LDLT status. In fact, in pediatric patients, the success rate of reaching anastomosis in the patients with LDLT after Kasai operation for biliary atresia and the others was 59.5% (25/42) and 86.7% (13/15) (*p* = 0.069), respectively. Although there is no significant difference between the two groups, prior history with multiple operations may be the factor responsible for the difficulty for BEA-ERC. In addition to the multiple operations, patients with biliary atresia had an intussusception antireflux valve, which hindered endoscopic insertion and increased the risk of resultant intestinal mucosal damage and perforation associated with balloon dilation.

The growth of the small intestine impeded endoscopic insertion. Weaver et al. [29] reported that the small intestine approximately doubles in length from birth to adulthood. Endoscopists must consider the growth of the small intestine during distal insertion in pediatric patients with surgically altered gastrointestinal anatomy.

Although the rate of adverse events in the pediatric patients who underwent BEA-ERC was similar to a previous report on ERCP [24], the present study had a significantly higher rate pertaining to adverse events in pediatric patients compared with adult patients, similar to the findings of Mercier et al. [24] in pediatric endoscopy. Prolonged procedures increase the risk of developing pancreatitis, while general anesthesia has an added risk for adverse events. To avoid keeping the patient in the prone position for a long period, the procedures were performed in the left lateral and in the semisupine positions, because some reports indicated that being in the prone position for a long period induced liver ischemia in pediatric patients [30,31]. None of the pediatric patients had fatal complications in this study. At our institution, we performed several trials to avoid adverse events. Most importantly, we ensured that only highly experienced clinicians performed the pediatric BEA-ERC procedures.

To avoid the increase in the pressure in the afferent limb, we inserted the DBE in clear water without the introduction of air. In addition, to avoid keeping the patient in the prone position for a long period, procedures were performed in the left lateral position and then in the semisupine position. Pediatric patients with biliary disorders and surgically altered gastrointestinal anatomy may require multiple endoscopic treatment sessions throughout their lives. BEA-ERC may be relatively minimally invasive and safe for biliary tract disorders, and concurrently maintain cosmetic outcomes. Therefore, the prevention of adhesions during surgery and reconstruction of the afferent limb after growth should also be considered. In our patients, partial resection of the small intestine for reconstruction of the afferent limb was sometimes performed in pediatric patients wherein endoscopic insertion was difficult owing to intestinal growth. Subsequently, endoscopic treatment became easier. For pediatric patients, surgery in conjunction with the use of continuous endoscopic treatment is also required (Figure 4). Though the efficacy of these trials could not be established, factors that ought to be considered to avoid adverse events in pediatric patients who underwent BEA-ERC have been presented.

Pediatric endoscopy is essential in patients wherein it is indicated [18,21]. Although the clinical success rate in pediatric BEA-ERC was lower compared with adults, it improved significantly (97.3%) in patients wherein the anastomotic site was reached. Thus, better candidates should be selected for pediatric BEA-ERC. Methods of determining the presence of adhesions before the endoscopic procedure should be developed in future studies.

This study is associated with some limitations. First, it used a retrospective single-center design that examined only a small number of patients. A prospective multicenter study involving a larger number of patients should be conducted to conclusively establish the efficacy and safety of pediatric BEA-ERC. Second, the conduct of DBE in BEA-ERC varied during the study period, specifically from long to short types.

## 5. Conclusions

BEA-ERC for biliary disorders in pediatric patients with surgically altered gastrointestinal anatomy is more difficult compared with adults. However, in patients wherein the balloon enteroscope was inserted to the level of the anastomosis, clinical success comparable to that in adult patients can be achieved. BEA-ERC in pediatric patients with biliary disorders should be considered as a viable treatment option.

## Figures and Tables

**Figure 1 jcm-10-03936-f001:**
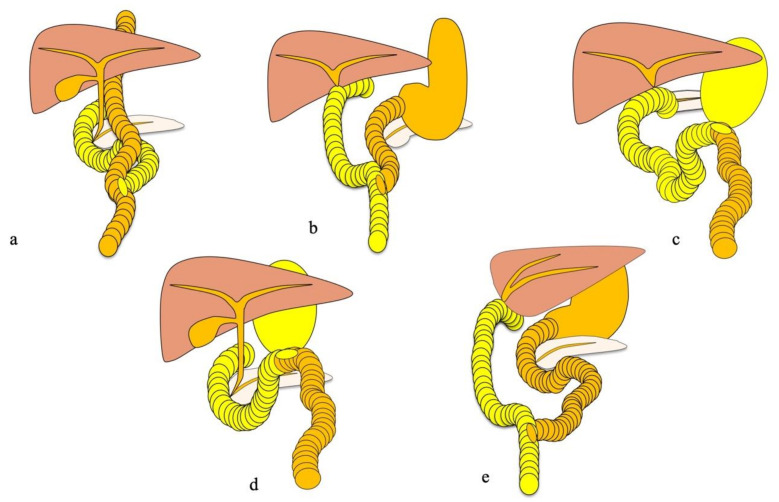
Operative procedures: (**a**) Roux-en-Y hepaticojejunostomy; (**b**) status post-living donor liver transplantation; (**c**) Roux-en-Y gastrectomy; (**d**) Billroth-II remnant resection; (**e**) pancreaticoduodenectomy.

**Figure 2 jcm-10-03936-f002:**
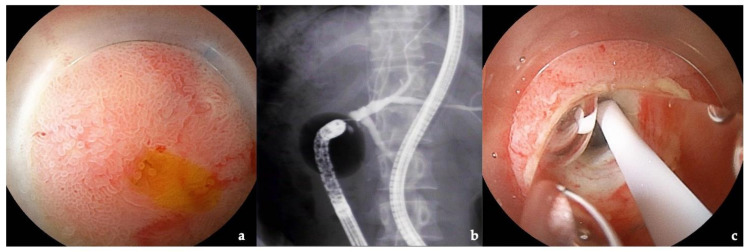
Patients for treatment of stenosis of the anastomosis using balloon enteroscopy-assisted endoscopic retrograde cholangiography: (**a**) status post-living donor liver transplant patient with biliary atresia: balloon enteroscopy-assisted endoscopic retrograde cholangiography showed the anastomosis with a stricture; (**b**) intrahepatic bile duct of the patient was dilated due to a stricture; (**c**) dilation with a balloon catheter.

**Figure 3 jcm-10-03936-f003:**
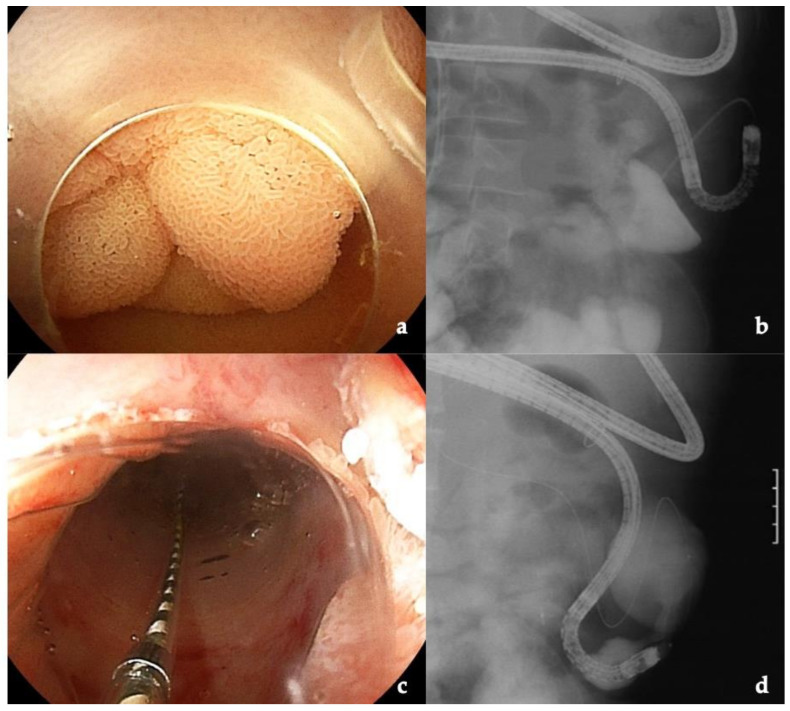
Intussusception antireflux valve in the afferent limb. The antireflux valve was created by invaginating the small intestine of the afferent limb to prevent cholangitis during childhood prior to living donor liver transplantation: (**a**) patient with intussusception antireflux valve; (**b**) antireflux valve with a stricture; (**c**) dilation with a balloon catheter; (**d**) the dilated antireflux valve.

**Figure 4 jcm-10-03936-f004:**
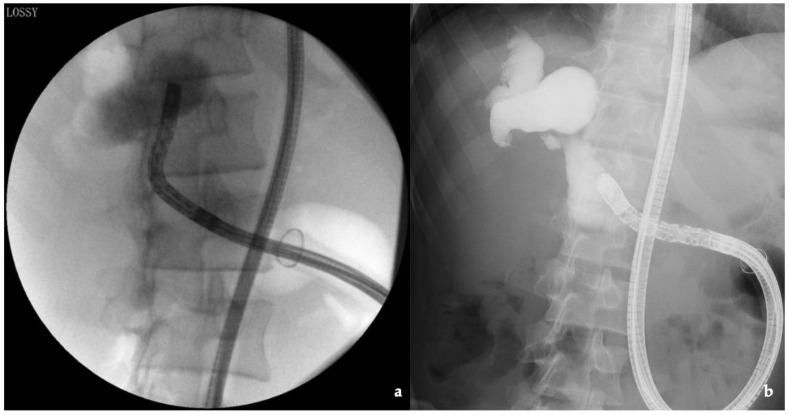
Reconstruction of the afferent limb to facilitate endoscopic insertion: (**a**) before reoperation, the insertion time required was approximately 70 min; (**b**) after reoperation, the insertion time was shortened to 22 min.

**Table 1 jcm-10-03936-t001:** Clinical profiles of enrolled patients.

Label	Pediatric Patients (*n* = 57, 127 Sessions)	Adult Patients (*n* = 291, 553 Sessions)	*p*-Value
male/female ratio (*n*)	24:33	195:96	<0.01
age (year), average ± standard deviation (SD)	10.0 ± 4.3 (min 3, max 18)	65.9 ± 17.3 (min 19, max 95)	<0.01
operative procedure (*n* (%))			<0.01
LDLT ^1^	53 (93.0)	32 (11.0)	
RYHJ	4 (7.0)	72 (24.7)	
RYG	0 (0)	102 (35.1)	
B-II	0 (0)	36 (12.4)	
PD	0 (0)	49 (16.8)	
indication for BEA-ERC, *n* (%)			<0.01
cholangitis, *n* (%)	32 (56.1)	32 (11.0)	
bile duct dilatation	20 (35.1)	80 (27.5)	
intrahepatic stone	3 (5.3)	46 (15.8)	
CBD stone	0 (0)	122 (41.9)	
others	2 (3.5)	9 (3.1)	
anesthesia, *n* (%)			<0.01
general	44 (77.2)	9 (3.1)	
intravenous	13 (22.8)	282 (96.9)	

^1^ B-II, Billroth-II remnant resection; BEA-ERC, balloon enteroscopy-assisted endoscopic retrograde cholangiography; LDLT, living donor liver transplantation; PD, pancreaticoduodenectomy; RYG, Roux-en-Y gastrectomy; RYHJ, Roux-en-Y hepaticojejunostomy.

**Table 2 jcm-10-03936-t002:** Primary indication for surgery.

Pediatric Patients (*n* = 57)			Adult Patients (*n* = 291)		
LDLT ^1^, *n* (%)	biliary atresia	42	LDLT	biliary atresia	15
	OTC deficiency	4		liver cirrhosis	2
	acute hepatitis	2		primary sclerosing cholangitis	2
	others	5		hepatocellular carcinoma	2
				others	11
RYHJ	biliary atresia	1	RYHJ	biliary atresia	6
	congenital bile duct dilatation	3		congenital bile duct dilatation	15
				stone	19
				biliary cancer	14
				pancreaticobiliary malfunction	5
				others	12
RYG		0	RYG	gastric cancer	88
				gastric tumor	1
				malignant lymphoma	1
				ulcer	3
				others	9
PD		0	PD	pancreatic tumor	30
				biliary tumor	8
				cancer of papilla of Vater	5
				others	6
B-Ⅱ		0	B-Ⅱ	ulcer	27
				gastric cancer	7
				others	2

^1^ B-II, Billroth-II remnant resection; BEA-ERC, balloon enteroscopy-assisted endoscopic retrograde cholangiography; IQR, interquartile range; LDLT, living donor liver transplantation; OTC, ornithine transcarbamylase; PD, pancreaticoduodenectomy; RYG, Roux-en-Y gastrectomy; RYHJ, Roux-en-Y hepaticojejunostomy.

**Table 3 jcm-10-03936-t003:** BEA-ERC for patients with surgically altered anatomy.

Label	Surgical Procedure	Pediatric Patients (*n* = 57) *n* (%)	Adult Patients (*n* = 291) *n* (%)	*p*-Value
success of reaching target site	LDLT ^1^	34 (62.7)	25 (78.1)	
	RYHJ	4 (100)	58 (80.5)	
	RYG	0 (0)	92 (90.1)	
	B-II	0 (0)	33 (91.7)	
	PD	0 (0)	48 (98.0)	
	total	38 (66.7)	256 (88.0)	<0.01
clinical success	LDLT	33 (64.9)	25 (78.1)	
	RYHJ	4 (100)	57 (79.1)	
	RYG	0 (0)	79 (77.5)	
	B-II	0 (0)	27 (75.0)	
	PD	0 (0)	46 (93.8)	
	total	37 (64.9)	234 (80.4)	0.014
time to reaching target site (min ± SD)		54.3 ± 32.8	40.6 ± 28.6	<0.01

^1^ B-II, Billroth-II remnant resection; BEA-ERC, balloon enteroscopy-assisted endoscopic retrograde cholangiography; SD, standard deviation; IQR, interquartile range; LDLT, living donor liver transplantation; PD, pancreaticoduodenectomy; RYG, Roux-en-Y gastrectomy; RYHJ, Roux-en Y hepaticojejunostomy.

**Table 4 jcm-10-03936-t004:** Success of reaching target site of BEA-ERC ^1^ for pediatric patients with biliary atresia after LDLT.

Label	Ages		*p*-Value
	≤12 y.o (*n* = 30)	>13 y.o (*n* = 13)	
Success of reaching target site (*n* (%))	18 (60.0)	8 (61.5)	0.743

^1^ BEA-ERC, balloon enteroscopy-assisted endoscopic retrograde cholangiography; LDLT, living donor liver transplantation.

**Table 5 jcm-10-03936-t005:** Adverse events caused by balloon enteroscopy-assisted endoscopic retrograde cholangiography.

Label	Label	Pediatric Patients (127 Sessions)	Adult Patients (553 Sessions)	*p*-Value
adverse events, *n* (%)	cholangitis	3 (2.3)	6 (1.1)	
	pancreatitis	3 (2.3)	25 (4.5)	
	bile duct injury	2 (1.5)	4 (0.7)	
	mucosal damage	3 (2.3)	2 (0.5)	
	pneumonia	0 (0)	1 (0.3)	
	fever	4 (3.1)	0 (0)	
	others	3 (2.3)	5 (0.9)	
	total	18 (14.2)	43 (7.7)	0.037
occurrence in each operative procedure, *n* (%)	LDLT ^1^	15 (11.8)	6 (1.1)	
	RYHJ	3 (2.4)	7 (1.3)	
	RYG	0 (0)	15 (2.7)	
	B-Ⅱ	0 (0)	9 (1.6)	
	PD	0 (0)	5 (0.9)	

^1^*B**-II*, Billroth-II remnant resection; *BEA-ERC*, balloon enteroscopy-assisted endoscopic retrograde cholangiography; *LDLT*, living donor liver transplantation; *PD*, pancreaticoduodenectomy; *RYG*, Roux-en-Y gastrectomy; *RYHJ*, Roux-en Y hepaticojejunostomy.

## Data Availability

The data that support the findings of this study available from the corresponding author, [A.K.], upon reasonable request.

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
