# Peer review of "The Efficacy and Safety of Balloon Enteroscopy-Assisted Endoscopic Retrograde Cholangiography in Pediatric Patients with Surgically Altered Gastrointestinal Anatomy"

_jcm, 2021, doi:10.3390/jcm10173936_

Round 1

Reviewer 1 Report

General comments

This article investigated the efficacy and safety of balloon enteroscopy-assisted endoscopic retrograde cholangiography (BEA-ERC) in pediatric patients with surgically altered gastrointestinal anatomy. BEA-ERC is less invasive and more useful than surgical reoperation for patients requiring biliary drainage, not only pediatric patients.

This study is valuable because it adequately describes not only the usefulness of BEA-ERC in pediatric patients but also and future issues.

  1. In this study, two types of a short-type double-balloon endoscope were used for BEA-ERC. It is possible that the difference in scope may affect the success rate of the procedure. I think it would be better to describe it in more detail, at least as a limitation of this study.
  2. Adult patient group had complications of post ERCP pancreatitis (PEP). Have any medications (e.g., NSAIDs) been administered to prevent pancreatitis? If prophylaxis medicine was given, please describe the protocol.
  3. BEA-ERC is generally considered to be a difficult procedure, and the success rate of that is influenced by the number of years of experience, especially in balloon endoscopy (BE). Please give additional consideration to this point if possible. Alternatively, please provide more details of the expert endoscopist in 2-2.
  4. Please add the minimum and maximum patient age in Table 1.
  5. In discussion section, you mentioned that undergoing multiple operations influenced intraabdominal adhesions. In this study, did the number of multiple operations affect the success rate of BEA-ERC?

Author Response

To Reviewer 1:

  1. In this study, two types of a short-type double-balloon endoscope were used for BEA-ERC. It is possible that the difference in scope may affect the success rate of the procedure. I think it would be better to describe it in more detail, at least as a limitation of this study.

- Thank you for this comment. We have changed the scope to EI-580BT with a large forceps channel since 2016. We could not find the differences of the clinical success between the 2.8 mm and 3.2 cm channels in pediatric and adult cases. As per your suggestion, we have incorporate a relevant description in the Results section on page 7, lines 158–160.

  1. Adult patient group had complications of post ERCP pancreatitis (PEP). Have any medications (e.g., NSAIDs) been administered to prevent pancreatitis? If prophylaxis medicine was given, please describe the protocol.

- As per your suggestion, we have inserted a relevant description in the Patients and Methods section on page 3 (line 83).

3.BEA-ERC is generally considered to be a difficult procedure, and the success rate of that is influenced by the number of years of experience, especially in balloon endoscopy (BE). Please give additional consideration to this point if possible. Alternatively, please provide more details of the expert endoscopist in 2-2.

- BEA-ERC was performed by DBE experts. As per your suggestion, we added the description in the Patients and Methods section on page 3 (lines 80–82).

  1. Please add the minimum and maximum patient age in Table 1.

- As per your suggestion, we added the minimum and maximum patient ages in Table 1.

  1. In discussion section, you mentioned that undergoing multiple operations influenced intraabdominal adhesions. In this study, did the number of multiple operations affect the success rate of BEA-ERC?

- In pediatric patients, the success rates of reaching anastomosis in the patients with LDLT after the Kasai operation for biliary atresia and for other procedures are 59.5% (25/42) and 86.7% (13/15) (p = 0.069), respectively. Although there is no significant difference between the two groups, the multiple operations may constitute the contributory factor associated with the difficulty for peforming the BEA-ERC procedure. As per your suggestion, we have added the description in the revised Discussion section (page 11, lines 225–229).

Reviewer 2 Report

As the authors declare in text the main limitations are the retrospective nature and the differences in the baseline characteristics between the two groups of patients. Despite this it represents a valid information source and the topic is very innovative. There are some remarks that I’d like to underline:

  • In the abstract the authors declare ”The procedure time was significantly lower in pediatric patients than in adults” but in the results section (line 152) the authors report that the procedure time was significantly longer in the same group. Please check.
  • The authors performed a sub-analysis for 12 years-old pediatric patients. Please summarize these results in a table.

Author Response

  1. In the abstract the authors declare ”The procedure time was significantly lower in pediatric patients than in adults” but in the results section (line 152) the authors report that the procedure time was significantly longer in the same group. Please check.

-Thank you for this comment. We apologize for the ambiguous description. We have changed the description in the Abstract section on page 1, lines 21–22.

  1. The authors performed a sub-analysis for 12 years-old pediatric patients. Please summarize these results in a table.

- As per your suggestion, we added a new Table 4 on pages 8–9. Given the addition of a new Table 4, the total number of tables regarding adverse events was changed to Table 5.
